# Why Do People Remain Attached to Unsafe Drinking Water Options? Quantitative Evidence from Southwestern Bangladesh

**Floris Loys Naus** [1],*, **Kennard Burer** [1], **Frank van Laerhoven** [1], **Jasper Griffioen** [1,2], **Kazi Matin Ahmed** [3] **and Paul Schot** [1]

1   Copernicus Institute of Sustainable Development, Utrecht University, Princetonlaan 8a, 3584 CB Utrecht, The Netherlands; Kennard.burer@gmail.com (K.B.); F.S.J.vanLaerhoven@uu.nl (F.v.L.); jasper.griffioen@tno.nl (J.G.); P.P.Schot@uu.nl (P.S.)
2   TNO Geological Survey of The Netherlands, Princetonlaan 6, 3584 CB Utrecht, The Netherlands
3   Department of Geology, Dhaka University, Dhaka 1000, Bangladesh; kmahmed@du.ac.bd
*   Correspondence: Florisnaus@uu.nl

**Abstract:** The acceptance of newly implemented, safe drinking water options is not guaranteed. In the Khulna and Satkhira districts, Bangladesh, pond water is pathogen-contaminated, while groundwater from shallow tubewells may be arsenic- or saline-contaminated. This study aims to determine why, as well as the extent to which, people are expected to remain attached to using these unsafe water options, compared to the following four safer drinking water options: deep tubewells, pond sand filters, vendor water, and rainwater harvesting. Through 262 surveys, this study explores whether five explanatory factors (risk, attitude, norms, reliability, and habit) pose barriers to switching from unsafe to safe drinking water options or whether they could act as facilitators of such a switch. Users' attachment to using pond water is generally low (facilitators: risk and attitude. Barrier: norms). Users are more attached to shallow tubewells (no facilitators. Barriers: reliability and habit). The safe alternatives (deep tubewell, rain water harvesting, pond sand filter, and vendor water) score significantly better than pond water and are estimated to have the potential to be adopted by pond water users. Deep tubewell, rain water harvesting, and pond sand filter also score better than shallow tubewells and could also have the potential to replace them. These findings may be used to optimise implementation strategies for safer drinking water alternatives.

**Keywords:** Bangladesh; rural drinking water supply; arsenic contamination; surface water health problems; attachment to unsafe drinking water

## 1. Introduction

In many cases around the world, it has been found that the acceptance of safe drinking water options varies and is not necessarily guaranteed [1–3]. To achieve the widespread adoption of safer, alternative drinking water options, the importance of so-called 'software activities' to actively promote a behavioural change among users of drinking water systems has been pointed out [4]. It is contended that when designing a strategy for providing alternative safe drinking water options, it is important to know what keeps people attached to their unsafe drinking water option and what could facilitate a switch from it.

In the Ganges–Brahmaputra–Meghna delta, with a population of over 170 million, unsafe drinking water is used due to drinking water resources being severely stressed. Surface water resources are polluted [5,6], meteorological water resources are subject to distinct seasonality [7,8], and shallow groundwater is often contaminated with arsenic [9–13]. In southwestern Bangladesh, salinity in surface

water and groundwater [9,14–16] puts further pressure on the available drinking water options, leading to the consumption of bacterially contaminated pond water [17] and of shallow groundwater with elevated levels of arsenic and salinity [18,19].

To overcome drinking water quality problems in southwestern Bangladesh, technical solutions have been introduced, often in a supply-driven manner. An example of a technical solution that has been piloted recently, albeit on a limited scale, is Managed Aquifer Recharge (MAR) [20]. Drinking water options that are considered relatively safe are scarce; they vary either spatially (i.e., deep tube wells (DTWs), pond sand filters (PSFs), and vendor water), or temporally (i.e., rainwater harvesting (RWH)). As a consequence, the users of unsafe ponds or shallow tube wells (STWs) do not always have safe alternatives. It should be noted that these alternative drinking water options are not always completely safe either, as DTW water is sometimes brackish or saline, PSFs do not always remove all the coliform bacteria [21,22], and the quality of vendor water cannot be guaranteed, as the source of vendor water is not always known [23] and the quality of RWH water can deteriorate over time [24,25]. Generally, though, these alternative options are assumed to be relatively safe, and in this paper the most problematic sources of water in terms of quality are considered to be ponds and STWs.

Similar to many other cases in the world, these safer alternative water options are not always adopted in Bangladesh. It was found that only 36% of arsenic-free drinking water options installed were functional [26]. Factors contributing to the likelihood of people adopting arsenic-free drinking water options have been extensively researched in Bangladesh [27–29]. However, so far, no research has focused on factors that cause people to stay attached to unsafe sources such as ponds and STWs. Information on users' attachment to unsafe water options makes it possible to perform ex-ante assessments of whether technical solutions will be successful, i.e., before costly implementation. This is especially valuable for designing a strategy for introducing technical solutions that will address the usage of pond or STW water in southwestern Bangladesh because safe alternatives are often unavailable here.

We set out to research users' attachment to unsafe drinking water options in the Khulna and Satkhira districts in southwestern Bangladesh, using explanatory factors identified from literature. Our aim was to assess factors that pose barriers to switching from unsafe to safe drinking water options and factors that will act as facilitators to such a switch. As a comparison, we investigated users' attachment to the most frequently available safe drinking water options, namely DTWs, PSFs, RWH, and vendor water. We conclude this paper by discussing the opportunities for, and limitations of, the provision of safe drinking water alternatives to replace the unsafe options.

## 2. Variables Explaining Variation in User Attachment to Drinking Water Sources

Why do people stay attached to unsafe or less safe drinking water options? To identify relevant explanatory variables that could help to answer this question, we consulted the literature explaining variation in the use of water and sanitation systems in developing countries. This literature unequivocally shows that people's drinking water choices result from their individual evaluation of a variety of factors [2–4].

The RANAS model [30] clusters factors thought to affect people's propensity to adopt a water source—e.g., one that is safe—in five separate blocks: **R**isk, **A**ttitudinal, **N**ormative, **A**bility and **S**elf-regulation (RANAS) factors. Risk factors describe people's perceived risk of falling sick from drinking the water from their water source. This is related to perceived vulnerability and the expected health effects associated with their drinking water options. The assumption is that people prefer the options that, in their view, pose a lower health risk. Attitude factors describe how people feel about their drinking water option, e.g., how they perceive the water's palatability, the effort of obtaining the water (i.e., collection time), or the price of the water. Here, the assumption is that people prefer options that provide tastier water, require less effort, and are less costly. Norm factors are related to what is perceived to be approved or disapproved of in their immediate social circle, e.g., whether the people who matter to them use similar or different drinking water options. The assumption here is

that people prefer options that are being used by most or all of the households around them. Ability factors describe whether people believe they are able to use a drinking water source, and whether they are confident of continuing to do so. The more this is the case, the more likely it is that people will use this option. Self-regulation factors describe the extent to which users can regulate their own behavior—e.g., to switch to a new drinking water option. People who are more confident in this regard can be expected to switch from an unsafe to a safe drinking water option more readily. It is important to realise that what counts is the perception of people. Whether this perception coincides with reality is of less importance.

So far, the explanatory variables proposed by the RANAS model have been used to explain variation in the adoption of a new source of safe drinking water [31–33]. However, we adopted the RANAS model as a basis to estimate people's attachment to their current safe or unsafe drinking water option. For this purpose, we had to adjust the model. Since we aimed to research the drinking water option that people already use, we assumed that our respondents were able to use this option. So, we removed ability factors from our adapted version of the RANAS model, replacing them with people's perception of the reliability of their current drinking water option. Additionally, the users in this study did not always have a safe alternative, so there was no clear target behaviour that the users needed to self-regulate. Therefore, instead of assessing self-regulation, we assessed whether people think they use their current option(s) out of habit. Our adapted version of the RANAS model—and the operationalisation thereof—is shown in Table 1.

**Table 1.** Operationalisation of the variables explaining the variation in people's attachment to their current drinking water source.

| Explanatory Factors | Definition | Interview Questions |
|---|---|---|
| **Risk** | | |
| Vulnerability | Risk of arsenic | How high or low do you think is the risk that you will develop arsenicosis? High risk = 1; Some risk = 2; Neutral = 3; No risk = 4 |
| | Health risk | How healthy do you think your drinking water is? Very unhealthy = 1; Unhealthy = 2; Neutral = 3; Healthy = 4; Very healthy = 5 |
| **Attitudes** | | |
| Instrumental beliefs | Collection Time | How long does it take in minutes to collect the water from the moment you leave the house until you come back (including walking, queuing, collecting)? Very short (<5 min) = 5; Short (5–9 min) = 4; medium (10–29 min) = 3; Long (30–60 min) = 2; Very long (>60 min) = 1 |
| | Cost | How do you feel about the cost of your water? Expensive = 1; Cheap = 2; Free = 3 |
| Affective beliefs | Palatability | How much do you like or dislike the taste of your drinking water? Strongly dislike = 1; Dislike = 2; Neutral = 3; Like = 4; Strongly like = 5 |
| **Norms** | | |
| Injunctive norm | Neighbours' opinion | Do your neighbours approve or disapprove of your drinking water source? Strongly disapprove = 1; Disapprove = 2; Neutral = 3; Approve = 4; Strongly approve = 5 |
| Descriptive norm | Regular convention | How many people from your community get water from your drinking water source? Few people/less than 10 = 1; Intermediate amount of people/between 10 and 100 = 2; Many people/more than 100 = 3 |
| **Reliability** | | |
| | Reliability | Will you be able to get water from your drinking water option in a month's time? Very unsure = 1; Unsure = 2; Neutral = 3; Sure = 4; Very sure = 5 |
| **Habit** | | |
| | Habit | Do you use your drinking water option out of habit? Very unsure = 1; Unsure = 2; Neutral = 3; Sure = 4; Very sure = 5 |

## 3. Methods

### 3.1. Sample Selection

Users' attachment to the two unsafe drinking water options (ponds and STWs) and the four most frequently available safe drinking water options (DTWs, PSFs, RWH, and vendor water) was assessed using surveys during a field campaign in January and February 2018 in multiple rural communities throughout the Khulna and Satkhira districts in southwestern Bangladesh. The data was collected by means of a face-to-face survey from 180 local community members who were using the safe and unsafe drinking water options that we were investigating. We are confident that our assumption that shallow groundwater and pond water are unsafe in our study area is justified because in our study region, the bacterial contamination of ponds is a known risk [17] and the consumption of arsenic- and salinity-contaminated water from STWs has been reported [18,19]. We based our choice of research location partly on regional maps of arsenic contamination [12,13] and salinity contamination [15]. The salinity of the STWs was further confirmed by measuring the Electrical Conductivity (EC) of the water. In addition, we aimed to obtain good coverage of the various different water options throughout the Khulna and Satkhira districts (Figure 1).

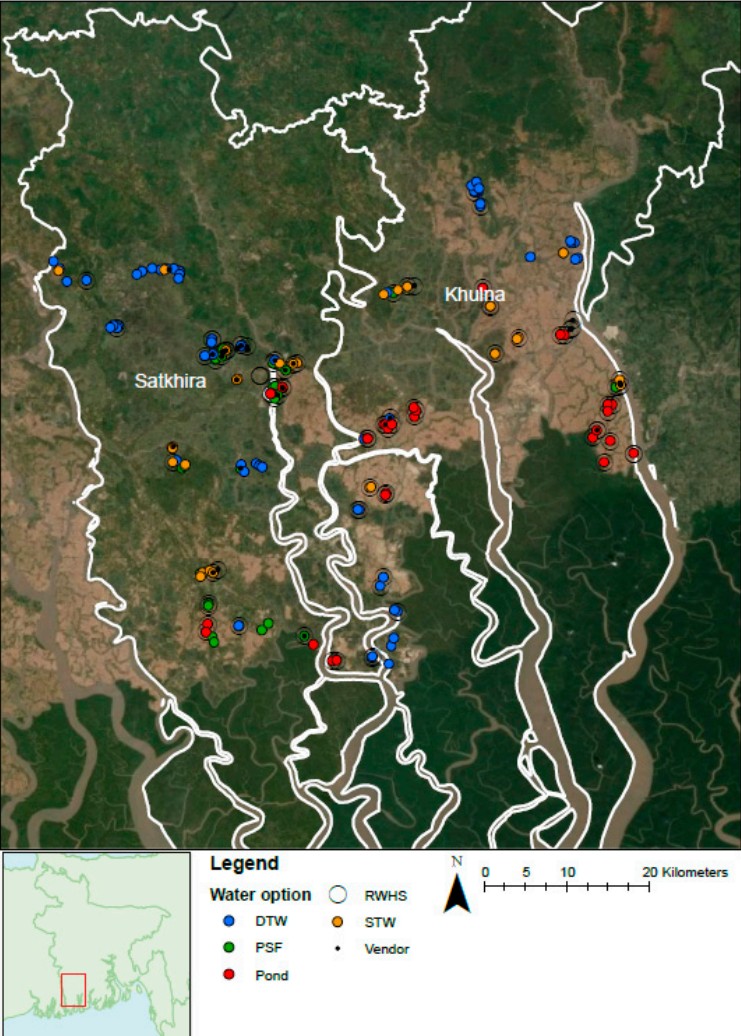

**Figure 1.** Locations of the surveys throughout the region. The symbols and colours indicate the water options related to the survey. DTW = deep tubewell; PSF = pond sand filter; RWHS = rainwater harvesting system; STW = shallow tubewell.

In each community, at least two members were selected for interview, using a random route method. In communities with multiple drinking water options available, more interviews were conducted. Figure 1 shows the locations of the interviews. If an interviewee used multiple drinking water options, we aimed to complete a separate questionnaire for each option; this resulted in a total of 262 completed questionnaires. The questionnaire questions were read out in Bengali several times; an interview took between 15 and 40 min, depending on the number of drinking water options the interviewee used. By keeping the length of the questionnaire, on average, around half an hour, we secured the participation of nearly all the households we approached, even though the interviews were unannounced.

*3.2. Operationalisation of the Explanatory Variables*

Both our dependent variable (i.e., drinking water source(s) used) and explanatory variables (i.e., perceived risk, attitude, norms, reliability, and habit) were measured through a structured questionnaire. The full questionnaire can be found in Appendix A. At the start of each interview, before administering the questionnaire, we recorded general information: the number of drinking water options the person used, the GPS location (Figure 1), the Electrical Conductivity (EC) of the drinking water option, the monetary costs of the drinking water option, and demographic data on the respondent. Next, the interviewee's perception of the explanatory factors was assessed, using the questions shown in Table 1. In Section 2, we explained how we identified the factors that might explain the variation in user attachment to drinking water sources. To an important extent, these factors are based on the ground-breaking work by Mosler [30]. For the factor 'risk', we asked about the interviewee's perception of the potability of the water and their perception of the risks associated with arsenicosis. The factor 'attitude' was assessed by the perception of cost, collection time, and palatability. To assess the factor 'norms', we asked what the interviewee thought the neighbours thought about the interviewee's drinking water option, and we asked the interviewee to estimate the size of the community using that drinking water option. To gauge the perceived reliability of the drinking water option, we asked the interviewees whether they thought they would be able to use it in a month's time. Since we administered the questionnaires halfway through the dry season, 'a month's time' corresponded to later in the dry season, when we expected drinking water options would be more limited. Lastly, we asked whether interviewees thought they were using their current water source purely out of habit. For the measurement of most variables, we used a conventional Likert scale rating (with five options, including a neutral option), rather than a forced-choice (ipsative) format. The variables that are less explicitly asking for an opinion (e.g., costs and regular convention), are measured by means of a three-point scale. It has been established that the stability, predictive validity, and concurrent validity of cumulative scores from Likert-type items are independent of the number of scale points utilised [34]. Therefore, we are confident that the use of various scales has no consequence for the outcome of our analysis.

*3.3. Data Analysis*

The raw survey data are available as Supplementary Materials. Our analysis is based on the state-of-the-art procedures employed in previous applications of the RANAS model. RANAS has so far only been used to explain the variation in the adoption of new, safe water options. However, we use it to estimate people's attachment to their currently used water options. We therefore developed the following adjusted approach for our analysis.

To be able to compare the explanatory factors, the answers to the questions were normalised, with the lowest or worst perception assigned a value of 1, and the highest or best perception assigned a value of 5. Next, the scores for the questions related to each of the explanatory factors were averaged. The mean and the standard deviation of each answer within each of the drinking water options were calculated. The mean values of the answers reveal the degree of attachment that that particular explanatory factor causes, with higher mean values indicating great attachment and lower mean values

indicating little attachment. For the unsafe drinking water options, values with a mean above 4 were seen as possible barriers to change, while values with a mean lower than 3 were seen as possible facilitators of change. For the safe drinking water options, values with a mean higher than 4 were seen as possible opportunities for a switch to the safe drinking water options, while values with a mean below 3 were seen as hampering a switch to the safe drinking water options. The standard deviations indicate the range in the scores between the interviewees.

Because the questionnaires only investigated interviewees' current drinking water options, qualitative interpretation was required to determine the likelihood that interviewees would switch from unsafe to safer drinking water options. To assist in this interpretation, we tested whether the differences in explanatory factors between the drinking water options were significant. Significant differences between the factors of the different classes were tested using the Kruskal–Wallis test [35] and Dunn's test [36].

## 4. Results

### 4.1. General Data

Table 2 shows general data obtained by the survey. It can be seen that rather than committing exclusively to one drinking water option, households in Bangladesh often use a portfolio of sources that, in varying ways and to varying extents, satisfy one or more of the several preferences they have with regard to their drinking water [37]. For pond and STW water (the two unsafe drinking water options), there were, respectively, 30 and 33 completed questionnaires. The numbers of surveys completed for the alternative safe drinking water options, PSF and vendor water, were similar: 34 and 30, respectively. The sources for which there were the most completed questionnaires were DTWs (70) and RWH (66). DTW is most often the sole drinking water option (69%), followed by STW (44%) and PSF (35%). Almost never used as the sole drinking water option are RWH (1%), pond water (7%), and vendor water (7%).

**Table 2.** General data for each drinking water option (1 USD = 84.5 BDT on 4 July 2019). STW = shallow tubewell, DTW = deep tubewell, RWH = Rainwater harvesting.

| Option | No. of Completed Questionnaires | Percentage Sole Drinking Water Option | Multiple Drinking Water Options | Median Cost (BDT/L) | Average Cost (BDT/L) | Usage L/(Day, Person) |
|---|---|---|---|---|---|---|
| Pond | 30 | 7% | 93% | 0.006 | 0.335 | 3.1 |
| STW | 32 | 44% | 56% | 0 | 0.031 | 4.3 |
| DTW | 70 | 69% | 31% | 0 | 0.077 | 3.1 |
| RWH | 66 | 1% | 99% | 0 | 0 | 2.6 |
| PSF | 34 | 35% | 65% | 0.022 | 0.131 | 3.1 |
| Vendor | 30 | 7% | 93% | 0.833 | 0.975 | 2.5 |

The average amount of drinking water used is 3.2 L per person per day, with only slight differences between the drinking water options: amounts of vendor and rainwater are slightly less, and amount of STW water is slightly more (Table 2). These amounts are comparable to those reported in a previous study [38], which found the median amount of consumed water to be 3.35 L per person per day. The total water use, including water for cooking and cleaning, is surely larger than these reported amounts, as often there is a separate water option for these other household activities.

The costs vary between the drinking water options. DTW, RWH, and STW are cheap, with a median of 0 BDT per litre (BDT/L) (1 USD = 84.5 BDT on 4 July 2019) and an average below 0.1 BDT/L. Pond and PSF are slightly more expensive: on average between 0.1 BDT/L and 0.35 BDT/L. The costs of using pond water are for treatment using alum as a disinfectant [39]. Vendor water is the most expensive, with an average close to 1 BDT/L (Table 2). The responsibility for collecting water is most often shared by all members of the family (45%), followed by being a responsibility for women only

(25% of cases), and being a responsibility for men only (14% of cases). In the remaining cases, water was delivered to the home (10%) or collected by servants (4%). Our finding that the responsibility for water collection is most often shared by the members of the household contrasts with the common finding that water collection is mostly a task for the female population [40].

*4.2. Explanatory Factors*

The means and standard deviations of the explanatory factors are presented in Table 3 for each drinking water option. The scores for the separate questions are shown in Table A1 in Appendix B. In general, only a few facilitators were identified: they scored below the neutral value of 3. Barriers (a score higher than 4) are more common, with each drinking water option having at least one. The standard deviations are often relatively large, which indicates that a neutral value close to 3 will still be judged negatively by an appreciable number of people.

**Table 3.** Standardised mean values of the explanatory factors. For the unsafe drinking water options, the facilitators (values lower than 3) are highlighted in blue, while the barriers (values higher than 4) are highlighted in yellow. For the safe drinking water options, barriers (values lower than 3) are highlighted in orange, while opportunities (values higher than 4) are highlighted in teal. As a reminder, the operationalisation of the factors is summarised below the table.

| Explanatory Factor | Unsafe Drinking Water Options | | | | | | Safe Drinking Water Options | | | | | |
|---|---|---|---|---|---|---|---|---|---|---|---|---|
| | Pond, n = 30 | | STW, n = 32 | | PSF, n = 34 | | Vendor, n = 30 | | DTW, n = 70 | | RWH, n = 66 | |
| | Mean | STD | Mean | STD | Mean | STD | Mean | STD | Mean | STD | Mean | STD |
| **Risk** = | 2.28 | 1.06 | 3.28 | 0.98 | 3.69 | 0.95 | 3.87 | 0.69 | 3.89 | 0.95 | 4.47 | 0.63 |
| **Attitude** & | 2.59 | 0.66 | 3.81 | 0.71 | 3.21 | 0.82 | 2.86 | 0.73 | 3.95 | 0.76 | 4.81 | 0.33 |
| **Norms** ^ | 4.22 | 0.68 | 3.5 | 1.09 | 4.49 | 0.29 | 4.21 | 0.54 | 4.06 | 0.73 | 3.81 | 0.74 |
| **Reliability** # | 3.73 | 1.48 | 4.16 | 0.99 | 3.27 | 1.35 | 3.27 | 1.43 | 4.02 | 1.05 | 2.95 | 1.32 |
| **Habit** ~ | 3.27 | 1.31 | 4.23 | 0.82 | 3.91 | 0.97 | 3.39 | 1.29 | 4.3 | 0.76 | 4.57 | 0.83 |

= Risk of arsenic, Health risk, & Collection time, Cost, Palatability, ^ Neighbours' opinion, Regular convention # Reliability, ~ Habit.

For ponds, we identified one barrier and two facilitators for switching to an alternative to this source of drinking water. The barrier is the factor 'norms', with a value of 4.22, which shows that people drink the pond water because the community generally approves of the option and that many people in the community drink it. Risk and attitude are facilitators of a switch: both score less than 3 (Table 3). In more detail, people perceive the pond water to be unhealthy; remarkably, they perceive that they are at risk of arsenic poisoning when drinking pond water. Additionally, people do not like its taste, judge the collection time to be long, and perceive the pond water to be expensive (Table A1). The combination of the negative attitude towards the ponds, the negative perception of risk and the low percentage of people that use ponds as their sole drinking water option (Table 2) suggests that people drink pond water when other drinking water options become unavailable, and not because they want to drink it.

No facilitators were identified for STWs, and habit and reliability were identified as barriers for a switch away from them. People use the STW water out of habit and because it is reliably available throughout the year. The two barriers and the lack of facilitators indicate that it is generally difficult to get STW users to switch away from their water source. Risk is the least important explanatory factor, with a mean somewhat above the neutral value of 3, indicating that emphasising the risks associated with STW water might be a way to get people to switch to a safer source of drinking water. Compared to ponds, the only factor that is lower for STWs is norms: the probable reason is that most people usually have their own private STW for their exclusive use. When the STWs were grouped according to their salinity, slight differences in the explanatory factors were observed (Table 4). The brackish

STWs (>2 mS/cm) have a higher score for reliability than the fresh STWs (<2 mS/cm), while fresh STWs have a higher attitude score than the brackish STWs (Table 4).

**Table 4.** Differences in explanatory factors between brackish and fresh shallow tubewells. Barriers (values higher than 4) are highlighted in yellow. The colours and symbols in the last column indicate significant differences and which group is higher: Brackish STW (teal +) or fresh STW (purple *). As a reminder, the operationalisation of the factors is summarised below the table.

| Explanatory Factor | Brackish STW | n = 13 | Fresh STW | n = 15 | Significance Dunn's Test |
|---|---|---|---|---|---|
| | Mean | Std | Mean | Std | |
| Risk = | 3.19 | 1.03 | 3.34 | 0.97 | |
| Attitude & | 3.35 | 0.70 | 4.12 | 0.54 | * |
| Norms ^ | 3.62 | 0.87 | 3.42 | 1.23 | |
| Reliability # | 4.62 | 0.87 | 3.84 | 0.96 | + |
| Habit ~ | 4.42 | 1.00 | 4.11 | 0.68 | |

= Risk of arsenic, Health risk, & Collection time, Cost, Palatability, ^ Neighbours' opinion, Regular convention # Reliability, ~ Habit.

For PSFs, scores for 'norms' are higher than 4, indicating that, similar to the ponds, PSFs are approved by the community and many people in the community drink water from them. The PSFs score the lowest for attitude and reliability, which score only slightly higher than the neutral score of 3. Examining the score for attitude in more detail reveals that it is controlled by a negative score for collection time and cost (Table A1). For vendor water, attitude scores lower than 3 and norm scores are above 4. The negative attitude is mostly attributable to the costs of the vendor water (Table A1), which suggests a preference for using cheaper unsafe drinking water options when they are available. The explanatory factors reliability and habit are only somewhat higher than the neutral score of 3, which—together with the low percentage of people solely using vendor water (Table 3)—suggests that the use of vendor water is incidental, despite the high value of norms. For DTWs, almost all factors score higher than 4. Habit, norms, and reliability score just above 4, but attitude and risk score just below 4. These positive scores for all explanatory factors reflect the fact that DTW is often the sole drinking water option used (Table 2). For RWH, risk, attitude, and habit are higher than 4, while norms scores just under 4. Reliability scores negatively, below 3. This can be attributed to the large seasonal fluctuations in the availability of rainwater, which result in almost all the people who use rainwater being unable to use it year-round (Table 2).

*4.3. Significant Differences Found for Explanatory Factors*

The results of the Dunn's tests are presented in Table 5. The unsafe pond water has many factors that score significantly more negatively than the factors of the safe drinking water options. Compared to DTW and RWH, pond has significantly lower values for risk, attitude, and habit, which suggests that there are multiple explanatory factors that could facilitate a switch from pond to DTW or RWH when these options are available. Compared to PSF and vendor, only risk is significantly lower, suggesting that when focusing on the risks of consuming pond water, users will be most amenable to switching from pond water to PSF or vendor water. Pond water does score significantly higher for norms than RWH does, suggesting that a change from pond water to rainwater might encounter resistance from the community. Compared to the other unsafe water source, STW, attitude is significantly more negative for pond water, indicating that users are likely to switch from pond water to STW water because STW water is more convenient to use than pond water.

**Table 5.** Significant differences in explantory factors between the drinking water options. Empty cells indicate non-significant pairs. The colours and symbols indicate, for the significant pairs, which group is higher: the first (teal +) or the second (purple *).

| Drinking Water Options | Group1 | Group2 | Risk = | Attitude & | Norms ˆ | Reliability # | Habit ~ |
|---|---|---|---|---|---|---|---|
| | | | | *p*-Value for Each Explanatory Factor | | | |
| **Unsafe** | Pond | STW | | * | + | | |
| | Pond | PSF | * | | | | |
| | Pond | Vendor | * | | | | |
| | Pond | DTW | * | * | | | * |
| | Pond | RWH | * | * | + | | * |
| | STW | PSF | | | * | | |
| | STW | Vendor | | + | | | |
| | STW | DTW | | | | | |
| | STW | RWH | * | * | | + | |
| **Safe** | PSF | Vendor | | | | | |
| | PSF | DTW | | * | | | |
| | PSF | RWH | * | * | + | | * |
| | Vendor | DTW | | * | | | * |
| | Vendor | RWH | * | * | | | * |
| | DTW | RWH | * | * | | + | |

= Risk of arsenic, Health risk, & Collection time, Cost, Palatability, ˆ Neighbours' opinion, Regular convention
# Reliability, ~ Habit.

STW water has significantly lower scores for risk and attitude than RWH and a significantly lower score for norms than PSF. STW users are most likely to use RWH water because RWH is more convenient to use and because of the perceived risk associated with drinking STW water. However, the greater reliability of the STWs compared with RWH does hamper a complete switch from STW water to RWH water. PSF scores significantly higher than STW for the factor 'norms', suggesting that a switch from STW to PSF has the highest chance of occurring when the community aspect of using PSFs is emphasised. Compared to vendor water, STW scores significantly higher for attitude, suggesting that the convenience of STW could limit a possible switch to vendor water. There are no significant differences between STW and DTW water, which indicates that switching from STW to DTW may not be easy, but also that this switch is not necessarily hindered by any of the explanatory factors studied. It may, therefore, be possible to convince the community to switch from STWs to DTWs by 'software' interventions [4].

There are also some significant differences between the improved drinking water options. PSF and vendor water score lower than DTW for attitude and lower than RWH for risk, attitude, and habit, suggesting that DTW and RWH would also have social potential to be adopted by people who use vendor or PSF water.

## 5. Discussion

The aim of our paper was to assess the attachment of users to their currently used unsafe water options. This information is valuable for ex-ante assessments of whether technical solutions will be successful and, consequently, for strategies introducing technical solutions that will address the usage of pond or STW water in the Khulna and Satkhira districts in southwestern Bangladesh, especially because safe alternatives are often unavailable here. Here, we will first discuss the implications of our findings for the potential for replacing the unsafe drinking water options. Next, we will discuss the potential and limitations of the most frequently present alternative drinking water options to replace the unsafe drinking water options.

### 5.1. Improvement Strategies

#### 5.1.1. Pond

A total of 17% of our respondents indicated that they currently use pond water for drinking water purposes (Table 2). The vast majority of pond water users (93%) use it in combination with other sources. A small proportion (7%) report that this is the sole source supplying their drinking water needs (Table 2). How can pond water users be pulled away from this relatively unsafe source? Our results suggest that the low scores for risk and attitude (Table 3) could be exploited: pond water users are aware of the risks associated with drinking pond water, and it seems possible that they could be persuaded to switch given that the scores for the risks associated with the sources PSF, vendor, DTW, and RWHS are significantly lower (Table 5). Additionally, the low risk score suggests that pond users would also be amenable to switching to other safe lower-risk alternatives that we did not investigate. The significantly lower score for the attitude to pond water compared to the scores for the attitude to DTW and RWHS is further evidence that a switch to these options might be achieved (Table 5). The overall low score for attitude suggests there is great potential for pond water users to switch to a more palatable, less time-consuming or cheaper alternative drinking water option (Tables 3 and A1). The safe alternative options would not have to be superior in all the inconveniences responsible for the low scores for the attitude to pond water (Table A1). For example, the finding that users of pond water often pay for their water suggests that they are able and could be willing to pay for alternative, safe drinking water options too (Tables 2 and A1). The alternative water sources, therefore, do not necessarily have to be free. The same applies to the collection time: the time taken to collect water from ponds is long (Table A1), so alternative, safe drinking water options sited centrally could still be attractive to pond water users.

The relatively high score for norms for pond water (Table 3) could inhibit the switch away from pond water, especially as previous studies have found that social norms greatly influence which drinking water option is chosen [28,32,33,41,42]. When providing alternative, safe drinking water options, a campaign focusing on changing the social norms may be necessary to get people to adopt the new safe options [4]. In general, the findings that pond water users take a long time to collect their water, do not like the taste of the water, think that it is expensive to pay for the addition of alum (Table A1), and have a negative perception of the pond water potability (Table 3) suggest that many of them would be very willing and motivated to switch to safer water options, but currently have no option to do so.

#### 5.1.2. Shallow Groundwater

A total of 18% of our respondents indicate they currently obtain drinking water from STWs (Table 2). A total of 44% of the households using STWs report that this is their sole source of drinking water (Table 2). Our findings suggest that the following might help achieve a switch from STW to safer options. Given that the scores for the explanatory factor attitude are significantly higher for STW users than for pond water users and that, compared to pond, STW scores significantly lower less often than the safe water options (Table 5), switching to safe drinking water options would probably be more difficult to achieve for STW users than for pond users.

Our findings show that when PSF systems are introduced, the best strategy to achieve adoption is probably to emphasise and change the community norms (Table 5), whereas when RWH systems are introduced, the best strategy is probably to emphasise the health risks associated with STW use (Table 5). As the risk associated with STWs scores the lowest of all the explanatory factors (Table 3), emphasising the risk may also achieve the best adoption result for other safe alternatives not investigated in our study. This contention is supported by the finding that, elsewhere in Bangladesh, highlighting the risk of arsenic caused people to change their water source in 65% of cases [43]. Unfortunately, in that study, most of the people changed to new, untested STWs, which shows that achieving a switch to completely different alternative drinking water options is not straightforward; it could be hindered by

the availability of alternative safe drinking water options. For STW users to change their behaviour, there must be other safe options nearby [29].

Given our finding that the barriers responsible for STW users remaining attached to their water source are reliability and habit (Table 3), the alternative safe water source should also be reliably available. The positive score for habit could be the largest barrier for a switch away from using STW water. Habits are hardest to change [30] and could prevent people switching away from STW water, even if they are willing to switch. In conclusion, the lack of clear facilitators, together with the clear barriers of reliability and norms, suggests that it is difficult to tackle the water quality problems associated with STW water. Lastly, it should be noted that there are differences in scores between brackish STWs and fresh STWs: for fresh STWs, attitude is valued significantly higher, whereas for brackish STWs, reliability is valued significantly higher for brackish STWs (Table 4), suggesting that different implementation strategies are required, depending on the salinity of the STW.

### 5.2. Implementation Potential of the Investigated Alternatives

Aside from the potential based on the explanatory factors investigated, other factors might hamper a switch from a currently used unsafe source to alternative safe drinking water options. Here, we discuss the potential of the investigated alternatives to replace the unsafe drinking water options by taking other factors into account.

#### 5.2.1. Rainwater Harvesting (RWH)

A total of 37% of our respondents report that they use RWH to satisfy their drinking water needs (Table 2). The vast majority (99%) of RWH users use it in combination with other options (Table 2). Harvesting rainwater is cheap and easy but storing it for a longer period of time is a challenge. RWH scored significantly better than pond for risk, attitude, and habit and scored significantly higher than STW for risk and attitude (Table 5), suggesting there is great potential for RWH to be adopted by both pond and STW users, as was also found by a previous study [44]. However, it is only available cheaply in the wet season, July to October. Most pond water users, and some of the STW users, already drink rainwater in the wet season. To overcome the seasonality of rainwater availability, larger reservoirs could be constructed. These require a large financial investment, and the stored rainwater will be more susceptible to the deterioration of quality [24,25]. Nevertheless, rainwater is available throughout the region, so the potential to implement RWH systems is everywhere, provided there is space for the reservoirs. This also suggests that the need for other safe water sources is limited to the dry season. This is important for the provision of the other safe drinking water options. If other alternatives need to rely on the financial contributions of the users, income can only be expected in the dry season. It should be noted that, given that most DTW users do not use rainwater throughout the year (Table 2), other water sources could completely replace rainwater, provided they score well enough (Table 3).

#### 5.2.2. PSFs

A total of 19% of our respondents indicate they use PSF (Table 2). Of all PSF users, 35% of the households use it as their only source of drinking water (Table 2). Our findings suggest that PSFs are likely to be adopted by pond users if differences in the perceived risks are emphasised (Table 5). The PSFs can be constructed next to currently used ponds, which could facilitate the switch from pond to PSFs. However, as already noted, the current design of PSFs is unable to remove all coliform bacteria [21,22], so there is a need for improved filters. A second drawback is that the PSFs require an initial financial investment, as well as additional financial contributions for maintenance. As a consequence of this, many previously installed PSFs have been abandoned. In a study of arsenic mitigation technologies in southeastern Bangladesh [45], the levels of abandonment of PSFs were found to be 87%. Even though the PSFs score significantly better than STWs for norms (Table 5), PSFs are less easily adopted as an alternative to arsenic-contaminated STWs. Filter plants were

found to be abandoned within weeks of construction [46] and, when attempts were made to replace arsenic-contaminated STWs, only a moderate acceptance of PSFs was found [27].

### 5.2.3. DTWs

As many as 39% of our respondents indicate that they rely on DTWs to cover their drinking water needs (Table 2). Of all DTW users, 69% indicate that they use no other source (Table 2). The largely positive factors suggest that DTWs have great potential to replace unsafe drinking water options at locations where deep groundwater is of sufficient quality (Table 3). The significantly higher scores for risk, attitude, and habit suggest that DTWs have great potential to be adopted by pond users (Table 5). We found no factors preventing or facilitating the replacement of STWs by DTWs (Table 5). The high potential of DTWs was also found previously [27,31,46]. The main limitation for DTWs remains the spatial availability of deep fresh groundwater. To some extent, the spatial limitation can be overcome by using piped systems to distribute deep groundwater from locations with deep fresh groundwater throughout the region. These piped systems have been found to be well accepted [27,46] and therefore have potential to be accepted as an alternative to ponds or STWs. However, the initial investment and maintenance costs are even higher than for the construction of manually operated DTWs. Additionally, it should be noted that fresh groundwater recharge is limited [9,14], which means that the use of deep groundwater would probably be unsustainable. Nevertheless, the mining of deep groundwater could be a useful solution for the short term. Before large infrastructure investments are made, the availability and sustainability should be researched in detail.

### 5.2.4. Vendor Water

A total of 17% of our respondents rely on water vendors, but relatively few of them (7%) rely on them solely (Table 2). Given the significantly higher score for the factor risk (Table 5), vendor water has the potential to replace ponds if the differences in risk between vendor water and pond water are emphasised. The significantly lower score for attitude compared to STWs suggest that it is unlikely that vendor water can replace STWs. The vendor water market remains non-transparent, and throughout the region it is often unclear what the source of the vendor water is [23]. Additionally, the fact that vendor water is expensive shows that it cannot fully replace the use of unsafe water, as people who cannot afford to pay for it have no choice but to use the unsafe, but cheap or free, drinking water options.

## 6. Conclusions

We assessed the attachment of users of unsafe water to their current drinking water option. Pond users are not very attached to their drinking water option and therefore it would be relatively easy to get people to switch from drinking pond water to safer alternatives. Compared with pond water users, STW users are more attached to their water source, indicating that it is expected to be more difficult to stop people consuming groundwater contaminated by salinity or arsenic than to stop them consuming bacterially contaminated pond water. This difference in attachment implies that efforts to improve public health may be most effective when focusing on users of pond water.

The greatest chance of getting people to switch from pond water is expected to be to focus on the risk and inconvenience of drinking pond water. The largest potential for getting STW users to adopt alternative, safe options is to focus on the risk associated with drinking groundwater with elevated levels of arsenic or salinity. The safer alternatives DTW, RWH, PSF, and vendor water are all estimated to have some potential to be adopted by pond water users, while RWH, DTW, and PSF could also replace STWs. However, the alternatives require large financial investments to make them available throughout the year and throughout the administrative districts of Khulna and Satkhira in southwest Bangladesh.

This paper illustrates the usefulness of research on why people are attached to their unsafe drinking water options and on why people do not adopt safe alternatives. Insight into users' attachment to

unsafe drinking water options could be used for ex-ante assessments of the likelihood of successfully introducing a future technical solution and for determining which factors to focus on. Based on the insights we gained, we suggest that future research seeks to compare respondents' perceptions of the explanatory variables, with objective measurements of the same. Additionally, we suggest combining quantitative analysis with more qualitative interpretation, for example, by means of data collection through semi-structured interviews and open-questions.

**Supplementary Materials:** The raw survey data are available online at http://www.mdpi.com/2073-4441/12/2/342/s1.

**Author Contributions:** Conceptualization, F.L.N., K.B., F.v.L. and P.S.; Data curation, F.L.N. and K.B.; Formal analysis, F.L.N., K.B., F.v.L., J.G. and P.S.; Funding acquisition, J.G., K.M.A. and P.S.; Investigation, F.L.N. and K.B.; Methodology, F.L.N., K.B., F.v.L. and P.S.; Project administration, F.v.L. and K.M.A.; Supervision, J.G., K.M.A. and P.S.; Visualization, F.L.N.; Writing—original draft, F.L.N., F.v.L., J.G. and P.S.; Writing—review & editing, F.L.N., F.v.L., J.G. and P.S. All authors have read and agreed to the published version of the manuscript.

**Funding:** This work is part of the Delta-MAR project funded by the Urbanising Deltas of the World (UDW) programme of the Dutch Research Council (NWO-WOTRO) (Grant number: OND1357179).

**Acknowledgments:** We would like to acknowledge the help of all the staff from the Delta-MAR office in Khulna for their support during the fieldwork campaigns, particularly Abir Delwaruzzaman, who helped immensely with facilitating the fieldwork. Additionally, we would like to thank all MSc students that assisted during the fieldwork, particularly Mahfuz, Arif, Bijoy and Refrat from Dhaka University. Joy Burrough is acknowledged for editing a near-final version of the manuscript. Lastly, we would like to thank Stevan Wang as editor and anonymous reviewer 1, 2 and 3 for their time and useful comments that helped improve the manuscript.

**Conflicts of Interest:** The authors declare no conflict of interest.

**Compliance with Ethical Standards:** Prior to each interview, the explicit consent of the human participant was sought.

## Appendix A

**Questionnaire**

**General Observations**

I. Location of interview: House//Street//Other
II. Date of interview
III. Gender: M//F
IV. Land use: Aquaculture (shrimp)//homestead//Crop culture (rice)
V. Distance to paved road: far away//close by//at road
VI. Wealth: Very Poor//Poor//Medium//Rich//Very Rich
VII. GPS location:

**General Questions**

1 What is your name?
2 How old are you?
3 What is your highest education level?
4 What is the name of your village?
5 What ward is this?
6 What union is this?
7 What upzilla is this?
8 How many members live in your household?

**Availability**

9 From what source do you get drinking water?
10 If tubewell, depth in feet?
11 What is/are your reason(s) for using your drinking water option?
12 How much drinking water do you collect per day for one person (in litres)?

13 Do you use any other sources at the moment (in the dry season)?
14 What is/are your reason(s) for using this drinking water options?
15 Do you use any other sources in the wet season?
16 What is/are your reason(s) for using this drinking water options?
17 Who collects the drinking water?
18 **Collection Time**. How long does it take in minutes to collect the water from the moment you leave the house until coming back (including walking, queuing, collecting)?

1.　Very long (>60 min)
2.　Long (30–60 min)
3.　Medium (10–29 min)
4.　Short (5–9 min)
5.　Very short (<5 min)

19 **Palatability**. How much do you like or dislike the taste of your drinking water?

1.　Strongly dislike
2.　Dislike
3.　Neutral
4.　Like
5.　Strongly like

20 **Palatability**. What is/are your reason(s) for liking or disliking it?
21 **Palatability**. Is there an iron taste in the water? Yes/No
22 **Palatability**. How is the saline taste of the water?

1.　Very saline
2.　Saline
3.　Neutral
4.　Not saline/fresh
5.　Not saline at all/very fresh

23 **Cost**. Is your water free, or do you have to pay? Yes/No
24 **Cost** How do you feel about the cost?

1.　Expensive
2.　Cheap
3.　Free

25 **Cost** How much taka do you pay per litre?
26 **Health**. How healthy do you think your drinking water is?

1.　Very unhealthy
2.　Unhealthy
3.　Neutral
4.　Healthy
5.　Very healthy

27 **Health**. Why do you think it is healthy/unhealthy?
28 **Arsenic**. Do you know about arsenicosis? Yes/No
29 **Arsenic**. What are the symptoms of arsenicosis?
30 **Arsenic**. How high or low do you think is the risk that you will develop arsenicosis?

1.  Large risk
2.  Some Risk
3.  Neutral
4.  No risk

31 **Neighbours' opinion**. What do your neighbours say about your drinking water?

1.  Strongly disapprove
2.  Disapprove
3.  Neutral
4.  Approve
5.  Strongly approve

32 **Regular convention** How many people from your community get water from your drinking water source?

1.  Few people/less than 10
2.  Intermediate amount of people/between 10 and 100
3.  Many people/more than 100

33 **Habit**. Do you use your drinking water option out of habit?

1.  Very unsure
2.  Unsure
3.  Neutral
4.  Sure
5.  Very sure

34 **Habit**. How long have you been drinking the water from this source?
35 **Habit**. What did you use before?
36 **Reliability**. Do you think you can get water from this source next month?

1.  Very unsure
2.  Unsure
3.  Neutral
4.  Sure
5.  Very sure

37 **Reliability**. Does your water source sometimes break down or becomes unavailable?
38 **Reliability**. What do you do if the water source breaks down or is unavailable?
39 **Reliability**. How often does it break down per month?
40 Water test for salinity (Electrical Conductivity) and temperature.

## Appendix B

**Table A1.** Standardised mean values for the answers to the different questions. For the unsafe drinking water options, the facilitators (values lower than 3) are highlighted in blue, while the barriers (values higher than 4) are highlighted in yellow. For the safe drinking water options, barriers (values lower than 3) are highlighted in orange, while opportunities (values higher than 4) are highlighted in teal.

| Question | Unsafe Drinking Water Options | | | | | | Safe Drinking Water Options | | | | | |
| | Pond, n = 30 | | STW, n = 32 | | PSF, n = 34 | | Vendor, n = 30 | | DTW, n = 70 | | RWH, n = 66 | |
| | Mean | STD | Mean | STD | Mean | STD | Mean | STD | Mean | STD | Mean | STD |
| Arsenic risk | 2.88 | 1.41 | 3.78 | 1.25 | 4.00 | 1.35 | 4.12 | 0.83 | 3.83 | 1.22 | 4.65 | 0.69 |
| Health | 2.03 | 1.13 | 3.00 | 1.02 | 3.50 | 0.99 | 3.66 | 0.86 | 3.80 | 1.04 | 4.42 | 0.70 |
| Collection time | 2.63 | 1.22 | 3.87 | 1.26 | 2.91 | 1.40 | 3.31 | 1.62 | 3.70 | 1.36 | 5.00 | 0.00 |
| Cost | 2.48 | 1.63 | 3.83 | 1.47 | 2.88 | 1.34 | 1.36 | 0.78 | 4.07 | 1.44 | 4.88 | 0.70 |
| Palatability | 2.63 | 0.85 | 3.69 | 0.90 | 3.85 | 0.74 | 3.80 | 0.85 | 4.04 | 0.86 | 4.64 | 0.57 |
| Neighbours' opinion | 3.77 | 0.82 | 3.59 | 1.04 | 4.09 | 0.45 | 3.86 | 0.64 | 4.04 | 0.75 | 4.09 | 0.46 |
| Community size | 4.79 | 0.83 | 3.41 | 1.80 | 5 | 0 | 5 | 0 | 4.18 | 1.16 | 1.67 | 1.45 |
| Reliability | 3.73 | 1.48 | 4.16 | 0.99 | 3.27 | 1.35 | 3.27 | 1.43 | 4.07 | 1.06 | 2.95 | 1.32 |
| Habit | 3.27 | 1.31 | 4.23 | 0.82 | 3.91 | 0.97 | 3.39 | 1.29 | 4.36 | 0.73 | 4.57 | 0.83 |

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
