# Peer review of "Why Do People Remain Attached to Unsafe Drinking Water Options? Quantitative Evidence from Southwestern Bangladesh"

_water, doi:10.3390/w12020342_

Round 1

Reviewer 1 Report

Your paper is a well written with a clear study design.  Your methods appear sound, and you sought to answer a concise issue.  You clearly answered the question of what water sources the populace is more attached to and where, should resources be made available, which water sources the populace would more readily switch from. In your conclusion, consider expanding upon the economic implications a bit - people rarely drink unsafe water by choice, but because of lack of resources or knowledge.  Because you found a lack of attachment to pond water, the implication is focusing scarce public water-related financial resources on these populations may get the most buy-in, and therefore, be a more effective approach toward protecting public health.

Reviewer 2 Report

The manuscript deals with the important issue of discussing the opportunities for and limitations to the provision of safe drinking water options as alternatives for the unsafe drinking water option. The following suggestions should be referred to. Why the five-point scale in the case of the effective beliefs variable? And a three-point scale in case of cost? The authors should explain why they used the mentioned variables presented in Table 1. Are there any similar studies or literature where this was used and applied. It should be enclosed to show state-of-the-art of the existing procedures for the analysis presented in the paper.

Reviewer 3 Report

My general overview on the manuscript is positive.

However, there are some points that the authors could clarify and some suggestions that they could consider.

Abstract: Normally in abstract we do not use abbreviations like RWG etc. Or at least it is quite uncommon. 

15. why people are expected to remain attached..perhaps : tend to remain 18. this study researchers: perhaps: explore

31 and is not NECESSARILY guaranteed.

38 consumption of ..occurs; perhaps: unsafe drinking water is used

40 [7,8] and [9-13] but e.g. 39 [[5, 6] here comma; be systematic

52 the source of vendor water is not always known. Therefore it is NOT necessarily safe.

72 of safe drinking water (options) as alternatives for unsafe (drinking water) options [omit those in (..)]

89 water options are used by people who matter to them. please correct: people use water sources that matter to them , or alike

130 the participation of nearly all the households. UNCLEAR to me. do you mean household members?

Some self-critical points for the discussion before conclusions

- What could perhaps been made otherwise?

Should you have left space for open questions. In structured interviews you can easily miss something. Semi-structured with a few open-ended questions could have been useful Did you consider the criteria "distance to the source". Except for vendors this is probably important. See e.g. Reselling and vending water. JAWWA, 83, 6: 63-69 (case from Mozambique)

Editorial:

- make shorter paragraphs. e.g. pages 2, 3, 5, 9, 10, 11,

Good luck for finalising the useful paper.
